# Study Protocol for a Randomized Controlled Trial Investigating the Effects of the Daily Consumption of Ruminant Milk on Digestive Comfort and Nutrition in Older Women: The YUMMI Study

**DOI:** 10.3390/nu16234215

**Published:** 2024-12-06

**Authors:** Shien Ping Ong, Jody C. Miller, Warren C. McNabb, Richard B. Gearry, Lara M. Ware, Jane A. Mullaney, Karl Fraser, Joanne Hort, Simone B. Bayer, Chris M. A. Frampton, Nicole C. Roy

**Affiliations:** 1Department of Human Nutrition, University of Otago, Dunedin 9016, New Zealand; ping.ong@postgrad.otago.ac.nz (S.P.O.); lara.ware@otago.ac.nz (L.M.W.); 2Riddet Institute, Massey University, Palmerston North 4410, New Zealand; w.mcnabb@massey.ac.nz (W.C.M.); jane.mullaney@agresearch.co.nz (J.A.M.); karl.fraser@agresearch.co.nz (K.F.); j.hort@massey.ac.nz (J.H.); 3High-Value Nutrition National Science Challenge, Liggins Institute, Auckland 1023, New Zealand; richard.gearry@otago.ac.nz (R.B.G.); simone.bayer@otago.ac.nz (S.B.B.); 4Department of Medicine, University of Otago, Christchurch 8011, New Zealand; chris.frampton@otago.ac.nz; 5AgResearch Grasslands, Palmerston North 4442, New Zealand; 6Food Experience and Sensory Testing (Feast) Laboratory, Palmerston North 4442, New Zealand

**Keywords:** ruminant milk, digestive comfort, older women, dietary intervention, study protocol

## Abstract

Background: Age-related changes can lead to dietary insufficiency in older adults. The inclusion of high-quality, nutrient-dense foods such as ruminant milks can significantly improve health outcomes. However, many older adults worldwide do not meet daily milk intake recommendations because of digestive discomfort and health concerns. Ovine and caprine milks are increasingly popular for their perceived digestive and nutritional benefits. While preclinical studies suggest differences in milk digestion, human studies investigating acute postprandial responses remain inconclusive, and the impacts of sustained milk consumption remain uncertain. Objectives: Hence, we present a randomized controlled trial investigating how the sustained consumption of bovine, caprine, or ovine milk influences digestion, nutrition, and metabolism in older women. Methods: A total of 165 healthy older women were randomized to receive bovine, caprine, or ovine milk, or no milk, twice daily for 12 weeks. The primary outcome is the impact of milk consumption on digestive comfort assessed via the Gastrointestinal Syndrome Rating Scale (GSRS). Secondary outcomes include changes in nutrient intake, plasma amino acid and lipid appearance, bowel habits, the gut microbiota, cardiometabolic health, physical function, physical activity, sleep, mood, sensory perception, and emotional response. Conclusions: The findings could inform dietary recommendations for older women and facilitate the development of targeted functional food products.

## 1. Introduction

The world’s population is aging. By 2030, every sixth person in the world will be over the age of 60 [1]. This brings unprecedented public health challenges as increased longevity is not always accompanied by increased health span [2]. In fact, disorders arising in those over the age of 60 account for 23% of the world’s disease burden [3].

Aging is a complex process characterized by molecular, cellular, physiological, and functional changes throughout the body that contribute to age-related declines and disorders [4]. The older population often experiences changes in eating behavior because of reduced appetite, taste and smell alterations, and oro-dental impairments, all of which may contribute to inadequate food intake, poor diet quality and nutritional deficiency [5,6]. This condition, often known as the “anorexia of aging”, is a significant predictor of negative health outcomes such as frailty, falls, and increased morbidity and mortality [5,7]. Lifestyle factors such as diet and nutrition are key modifiable targets for preserving health and preventing or slowing the progression of age-related disorders [2,8].

Most dietary recommendations for the older population focus on the addition of nutrient-dense foods to address key nutritional issues and meet nutrient and caloric needs [6,9]. The nutrient richness of dairy products makes them well-suited for this paradigm. Adults meeting daily dairy recommendations are more likely to have adequate intakes of key nutrients such as protein, calcium, magnesium, zinc, riboflavin, and vitamin B12 [10,11,12,13,14], nutrients that are often low in the older population [15,16,17,18,19]. Despite this, the average adult consumes less than one serving of dairy each day [20], often because of concerns about digestive discomfort, weight gain, and other perceived adverse health effects of dairy consumption [21,22]. Women, in particular, tend to report more gastrointestinal symptoms [23,24,25,26,27], possibly because of an interplay of physiological and psychosocial factors [28,29], and generally consume less milk compared to men [11,21,30]. Older women are also at greater risk for nutrient insufficiency, though reasons for this remain unclear and may include physiological, socioeconomic, and societal factors such as gender inequality and widowhood [31,32,33].

Bovine milk and its derivatives are widely consumed because of their widespread availability and large production volumes [34]. However, changing dietary preferences, cow milk allergy, and lactose intolerance have led to a rising interest in alternative milks [35,36]. Other ruminant milks, such as caprine and ovine milks, are gaining popularity because of the purported health and environmental benefits they have over bovine milk [37,38]. Milk composition can vary considerably within species because of breed, age, season, diet, and lactation stage [39,40]. However, greater compositional, structural and physicochemical differences exist between milks from different species which may impact their digestive behavior and tolerability [34,39,40,41]. For instance, while bovine and caprine milks have similar fat and protein content, ovine milk is notably higher in both macronutrients [42]. Additionally, different proportions of casein and whey, the two major protein fractions in ruminant milks, could impact curd formation in the gastric environment and influence protein hydrolysis rates and digestive behavior [37,40,43,44]. Furthermore, the breakdown and absorption of milk fats in the stomach and small intestine may also be influenced by differences in fatty acid composition and milk fat globule size between species [34,45,46,47].

In vitro studies exploring the digestive dynamics of different ruminant milks [48,49,50], and human trials focusing on acute responses to consuming these milks [51,52,53,54,55] suggest potential differences in digestive behavior. However, there is a paucity of well-controlled trials, particularly in the older population, to understand the long-term effects of consuming different ruminant milks. Therefore, we designed the effects of daily consumption of ruminant milk on digestive comfort and nutrition in older women (YUMMI) study to address this research gap by evaluating the effects of consuming ruminant milks (i.e., bovine, caprine, or ovine) on digestive comfort, nutrition and metabolism in older New Zealand women over 12 weeks. We hypothesize that different milks may have different impacts on digestive comfort and that the addition of ruminant milks, regardless of species, could improve nutrient intake in older women.

The primary aim of the study is to determine the effects of 12-week ruminant milk consumption on digestive comfort of older women using the Gastrointestinal Syndrome Rating Scale (GSRS). The key secondary aim is to determine the effects of 12-week ruminant milk consumption on nutrient intake of older women based on biomarkers of nutrient status (amino acids and trace elements concentrations) and self-reported dietary data (protein intake). Exploratory outcomes include evaluating the effects of 12-week ruminant milk consumption on bowel habits (bowel movement diary), cardiometabolic health (blood lipids, hemoglobin A1c (HbA1c), body weight, body composition, blood pressure), physical function (timed walk test, handgrip strength test), physical activity (accelerometry), and sleep (accelerometry and Leeds Sleep Evaluation Questionnaire), mood (Patient-Reported Outcomes Measurement Information System (PROMIS): Anxiety, Depression), sensory and emotional responses (sensory and emotional assessments), and the composition and predictive function of the gut microbiota (metagenomic sequencing of stool samples).

In addition to the 12-week intervention described above, we also incorporated an investigation to determine the acute effects of consuming a bolus dose of ruminant milk on digestive comfort (appetite and satiety questionnaire), digestive function (amino acid appearance), and acute cardiometabolic effects (GlucoTRIG index [56]). This investigation was conducted both at baseline and again after the 12-week intervention in a sub-group of participants to study potential adaptations to nutrient metabolism with regular milk intake.

## 2. Materials and Methods

### 2.1. Study Design and Setting

The YUMMI study is a 12-week multi-arm, parallel, multi-center, single blinded randomized controlled trial carried out at the Dunedin and Christchurch campuses of the University of Otago, New Zealand. A total of 165 participants were randomized to one of four intervention groups: habitual diet supplemented with 500 mL of either bovine milk, caprine milk, or ovine milk daily for 12 weeks; or a habitual diet with no additional milk (control) group to study the effects of sustained consumption of these milks on digestive comfort, nutrient intake, and metabolism among older women. Bovine milk powder was supplied by Miraka Ltd. (Taupo, New Zealand), caprine milk powder by NIG Nutritionals Ltd. (Auckland, New Zealand), and ovine milk by Spring Sheep Milk Co. (Hamilton, New Zealand). After a lead-in period of 2 weeks, participants consumed their assigned milk, attended clinic visits for measurements and biological sample collection and completed validated questionnaires over the course of 12 weeks.

At baseline and after 12 weeks of the intervention, a subgroup of participants from each milk group consumed a bolus dose (500 mL) of their assigned milk, had sequential blood samples taken and completed assessments to assess acute postprandial plasma amino acids, triglycerides, insulin, and appetite hormones in response to consuming the study milks.

The trial protocol has been prospectively registered with the Australia and New Zealand clinical trials Registry (ANZCTR) with registration number ACTRN12622001161718. Data collection commenced in January 2023 and concluded on the 27th of March 2024 and lab analyses have commenced.

### 2.2. Recruitment and Eligibility Criteria

A total of 165 healthy older women were recruited from the general population in Dunedin and Christchurch through a combination of online and community-based recruitment strategies. This included study posters put up around the university, local general practitioners, local hospitals, and local churches, newspaper advertisements, radio advertisements, community talks and presentations to older adult groups (University of the Third Age, Probus, Age Concern Otago, 60 plus), social media posts (Facebook, Instagram) and through word of mouth by Māori and other health nurses. All individuals who expressed an interest in the study were invited for screening and assessed for eligibility according to the inclusion and exclusion criteria listed below:Inclusion criteria:
Healthy, non-institutionalized women aged 60 to 80 years.Body mass index >18.5 or <40 kg/m^2^.Do not undertake structured exercise for more than 2 h per day.
Exclusion criteria:
Unable to give informed consent.Unable or unwilling to comply with the study procedures.Have taken antibiotics within 4 weeks of starting the study.Have taken certain prescribed medications or recreational drugs that could affect the gastrointestinal tract within 4 weeks of starting the study: opioids, non-steroidal anti-inflammatory drugs, laxatives, prebiotic, or probiotic supplements.Have medical history of gastrointestinal surgery or disorders (inflammatory bowel disease, ulcerative colitis, coeliac disease, Crohn’s disease), cardiorespiratory problems, uncontrolled diabetes mellitus, bleeding disorders, sleep disorders, psychiatric conditions (major depressive disorder, schizophrenia, bipolar disorder, post-traumatic stress disorder).Have alarm features associated with significant gastrointestinal or other disorders, such as burning pain in the epigastrium, which increases during the night and wakes the patient up; frequent vomiting; loss of appetite; lower gastrointestinal bleeding; odynophagia; dysphagia; palpable abdominal mass; lymphadenopathy; and jaundice.Experienced unintentional weight loss of ≥5% within the month of starting the study.Have dairy intolerance or ruminant milk allergy.Have a high habitual milk intake (≥500 mL ≥4 days per week).Malnutrition Screening Tool score of 2 or more points.Use tobacco, including cigarette smoking or other use of tobacco or nicotine-containing products.Excessive alcohol intake, i.e., >20g of pure alcohol (2 drinks)/d on average (>21 standard drinks a week).Screening blood biochemistries:▪HbA1c > 54 mmol/mol Hb (appropriate for older adults) [57].▪Blood biochemistry test results (Table A1) > 2 times the upper limit of normal.▪Hb < 120 g/L.
Non-exclusion criteria:

Cardiovascular disease, hypertension, diabetes mellitus, or depression that are well-controlled with medical intervention.

Out of the 165 participants enrolled in the study, 9 participants allocated to the milk groups consented to and completed the acute sub-study.

### 2.3. Participant Flow

A flow diagram of the study protocol is outlined in Figure 1. Participants attended the study clinic for one screening visit and three study visits (week −2, week 0 and week 12) in addition to completing daily and weekly questionnaires on a mobile application (app) developed for the study or on paper (Table A2). Participants collected their 4-week supply of milk powder at week 0, week 4 and week 8 and returned any unused milk powder at week 4, week 8 and week 12.

#### 2.3.1. Screening

Respondents who expressed interest in the study completed an online pre-screening questionnaire to evaluate their current health status and lifestyle habits. This questionnaire consisted of selected question domains adapted from the New South Wales Population Health Survey (NSW Population Health Survey Questionnaires, Methods, Results and Data Dictionaries for Adults, n.d.), covering areas for diabetes, smoking, alcohol consumption, and physical activity. Potentially eligible respondents were invited to attend a screening visit at the study clinic where they had the opportunity to ask questions about the study and their responses to screening questions were reviewed by a research assistant. Once informed consent was obtained, respondents had their anthropometric measurements (height, weight) taken and provided fasted blood samples (for screening blood biochemistries, Table A1) to determine their study eligibility.

#### 2.3.2. Main Study

Once participants’ blood chemistry was confirmed to meet study requirements, eligible individuals were formally enrolled, assigned a unique study identifier, and began their 2-week lead-in phase. Participants received links to relevant questionnaires and surveys (Table A2) along with instructions on how to complete them. They were given the option of completing the questionnaires via the study app or on paper.

One week prior to their baseline visit (week −1) and on the last week of the study (week 11), participants collected three-day diet records on three randomly assigned days (2 weekdays, 1 weekend day). They also tracked their physical activity and sleep for 7 days using an accelerometer (ActiGraph wGT3X-BT, Actigraph LLC, Pensacola, FL, USA). Participants underwent a dual-energy X-ray absorptiometry (DEXA) scan one week before their baseline and final study visits. Dunedin participants were scanned on the Lunar iDXA scanner (GE HealthCare, Madison, Wisconsin, USA) while Christchurch participants were scanned on the Lunar Prodigy scanner (GE HealthCare, Madison, Wisconsin, USA),

One day prior to their baseline and final study visits, participants collected stool samples using the OMNIgene GUT collection systems OMR205 and ME200 (DNA Genotek Inc., Ottawa, ON, Canada). They kept the tubes at room temperature until delivery to the clinics the following day. The tubes were filled with proprietary stabilization solutions designed for preserving DNA, RNA and metabolome at room temperature for 30, 10, and 7 days, respectively. Additionally, participants consumed a standardized meal provided to them for their evening meal before their baseline and final study visits, and fasted thereafter for 10 h. The standardized meal consisted of a Plantry Foods Lasagne (350 g, Goodman Fielder, New South Wales, Australia) or a Weight Watchers Mushroom and Pumpkin Risotto (320 g, as a gluten-free option, WW International Inc., New York, NY, USA) and an Alpro Soya Chocolate Dessert (125 g, Alpro, Ghent, Belgium). The meal was provided to standardize nutrient intake among participants and mimimize variability in blood biomarkers because of dietary factors apart from the study intervention [58].

At their baseline and final study visits, participants had their blood pressure, fasted blood samples and anthropometric measurements (height, weight, waist circumference) taken. They also completed functional tests such as a timed walk test and a hand grip strength test. At baseline, participants filled out the Economic Living Standard Index Short Form (ELSISF) to measure their economic standard of living, i.e., the material aspect of wellbeing reflected in their consumption habits and personal possessions [59]. Data collected will be used to assess their socioeconomic status and identify any important issues that may confound study data.

Full details of study outcomes and measurements are provided in Section 3.

#### 2.3.3. Acute Sub-Study

Before the baseline visits, all participants allocated into milk groups were invited to take part in the acute sub-study. Those who consented consumed a bolus dose (500 mL) of their allocated milk and had sequential blood samples drawn for acute assessments at 0, 60, 120, 180, 240, and 300 min during their baseline and final visits. They also filled out a 100 mm Visual Analog Score (VAS) over the five hours to measure gastrointestinal symptoms and appetite parameters.

### 2.4. Randomization and Blinding

Participants were randomly assigned to one of four groups: bovine milk, caprine milk, ovine milk, and control (no milk) in a 1:1:1:1 ratio with a pre-specified computer-generated random block randomization list prepared by the study statistician. Block sizes of eight were used without any stratification. To mimize bias, participant allocation was carried out by the principal investigator, who supplied the allocation for each participant to the research team a week before the participants were scheduled to come in for their baseline visits and all data analysis will be performed by blinded researchers.

Study participants were not informed about their milk intervention, but no attempt was made to mask the flavor of the milks. Researchers assessing study outcomes and conducting data analysis will remain blinded to participation allocation until the completion of statistical analyses for the primary and key secondary outcomes.

### 2.5. Intervention

Participants in the milk groups were given a 4-week supply of their assigned milk powder at baseline (week 0), week 4 and week 8. They were asked to maintain their usual diet and consume one serving of their assigned milk twice daily, with breakfast and lunch, over the 12-week intervention period. Participants reconstituted their assigned milk powder by weighing out their daily portion (as detailed in Table 1) and dissolving it in lukewarm water before making up the milk to 500 mL and mixing it thoroughly. Once mixed, the milk was refrigerated. The following day, they would have half with breakfast and the other half with lunch. The daily portions were calculated to match the nutrient composition of fresh milk while providing the same volume of study milk to all participants in milk groups. Depending on the type of milk, the intervention would add 9–14 g of protein to each meal, totalling 18–29 g of protein each day. Participants were asked to consume their milk at breakfast and lunch to increase protein intakes during these meals, as protein intakes are typically low for these meals [60,61]. Participants in the control group were asked to follow their usual diet for 12 weeks with no restrictions on their milk or dairy intake.

### 2.6. Attrition and Compliance/Adherence

To reduce attrition, strategies such as creating a welcoming environment for the participants, flexibility in scheduling clinic visits (±2 days), sending out email reminders the day prior to each scheduled visit and providing financial remuneration to those taking part in the study were employed. For the latter, all possible participants who attended the screening clinic received a $20 supermarket voucher, and those participating in the study received a $100 supermarket voucher after the baseline visit and an additional $100 upon completion of the study.

To improve compliance, participants in the milk groups were given Vital Zing Caramel Milk Drops (Vital Zing, Auckland, New Zealand) to increase the palatability of the milks. Fortnightly phone calls to the participants were also made by study personnel to discuss adherence and troubleshoot any difficulties encountered. Participants struggling to adhere to the interventions were offered support in the form of counseling from a dietetics student, supervised by a registered dietitian, to help them improve compliance. To assess compliance, participants in the milk groups were asked to fill out a daily compliance diary, reminded not to share their milk powder with anyone else and asked to return any unused milk powder during their week 4, 8 and 12 clinic visits. To quantify milk consumption, the unused milk powder was weighed and the amount deducted from the total provided, with the understanding that errors or spills may have occurred.

### 2.7. Participation and Withdrawal from the Study

Participation in the study was voluntary, and participants had the right to withdraw from the study at any time. The principal investigator had the right to discontinue a participant from the study if they became ineligible, they experienced significant adverse events such as allergic reactions, there was significant protocol deviation or non-compliance with study requirements, or they withdrew their consent.

### 2.8. Risk and Adverse Events

While no significant adverse events have been reported for studies of this nature, and the study and the intervention are low risk, participants might experience events such as gastrointestinal discomfort and weight gain caused by the addition of milk, discomfort caused by venepuncture, and low blood sugar due to fasting. Participants were provided information about these risks before they provided informed consent for the study. Management actions were put in place such as referral to a dietetics student, supervised by a registered dietitian, for dietary advice and support, the engagement of trained phlebotomists, and the provision of snacks and beverages after blood draws.

Adverse events were monitored and recorded by the research team, and the research team assessed and managed the condition. In the event of significant adverse events, the participant would be referred to the study clinician for assessment and management plan, and the event recorded in the case report form.

## 3. Study Outcomes and Measurements

### 3.1. Weekly Questionnaires

Participants completed weekly questionnaires on the study app or on paper from baseline until the end of the 12-week intervention period. These weekly questionnaires are described briefly below.

#### 3.1.1. Gastrointestinal Syndrome Rating Scale (GSRS)

The primary outcomes are between-group differences in abdominal pain scores and bowel comfort scores from baseline to week 12, measured using the abdominal pain domain in the GSRS. Upon enrolment, participants filled out the GSRS on a weekly basis. The GSRS is a validated questionnaire that uses a 7-level Likert scale (1–7) to assess the frequency and intensity of gastrointestinal symptoms experienced in the past 7 days [63]. It contains 15 items divided into five domains covering the gastrointestinal system: Reflux, Abdominal pain, Indigestion, Diarrhea, and Constipation. A higher score represents more frequent and/or severe symptoms experienced by the patients. Significant responders to the intervention will be those reporting at least a 0.6 reduction in abdominal pain severity score, the minimal important difference [64], compared to baseline.

#### 3.1.2. Patient-Reported Outcomes Measurement Information System (PROMIS): Anxiety and Depression

PROMIS is a set of measures covering different domains that evaluate and monitor physical, mental, and social health [65]. Specific domains can be picked to be integrated into diverse data collection tools. In this case, two scales from the psychosocial domain, anxiety and depression [66], were used to evaluate psychosocial symptoms in the past seven days in detail. This evaluation allows the examination of potential associations or confounding with gastrointestinal symptoms or gut microbiota parameters.

#### 3.1.3. Leeds Sleep Evaluation Questionnaire (LSEQ)

LSEQ is an instrument used to monitor self-perceived changes in sleep which consists of 10,100 mm Visual Analog Scales [67]. The questionnaire was administered on paper. The questionnaire contains questions related to ease of initiating sleep, quality of sleep, ease of waking and behavior following wakefulness. This allows the study of the association between milk intake and sleep quality.

#### 3.1.4. Sensory and Emotional Assessments

After study allocation, participants in the milk groups completed a weekly sensory survey where they rated their overall liking and specific liking of the flavor, texture, and aftertaste of their allocated study milk on a nine-point hedonic scale [68]. A circumplex question in which they indicate the level of pleasantness and arousal they experience after having their study milk assessed their emotional response to their study milk [69]. These assessments provide insights into how participants perceived and experienced their study milk and its impacts on milk intake.

#### 3.1.5. Milk-Specific Food Frequency Questionnaire

Participants recorded the type and quantity of milk and milk-containing products (e.g., coffee, tea, and hot chocolate.) they consumed outside the study. This allows the assessment of habitual non-study milk intake.

### 3.2. Daily Questionnaires

Participants completed the following daily questionnaires on the study app or paper from baseline until the end of the 12-week intervention period.

#### 3.2.1. Bowel Movement Diary

Participants recorded the frequency, spontaneity and completeness of their bowel movements, ease of defecation or level of straining and stool form according to the Bristol Stool Scale. This provides a comprehensive record of their bowel habits and captures any changes that may have occurred throughout the study.

#### 3.2.2. Compliance Diary

After randomization, participants in the milk groups recorded their consumption of their assigned milk. This provides a comprehensive record of their compliance with study instructions and captures any changes that may have occurred throughout the study.

### 3.3. Dietary Assessment

Participants completed three-day weighed diet records [70] during the week prior to their baseline visit and the week prior to their final visit (week 12). The three recordings were randomly assigned (two weekdays and one weekend day). This will be used to assess baseline protein intake and the impact of increased ruminant milk consumption on protein intake. The data will also be used to determine associations between diet and other study outcomes.

Participants were given a set of kitchen scales, a photo booklet depicting different portion sizes, and verbal and written instructions on how to measure and record their diet. When they came in for their baseline and final visits, the research team reviewed their diet records to check whether adequate details have been provided and whether there were any issues with recording. The diet record data will be entered into FoodWorks11 (Xyris, Brisbane, Australia) by a nutrition student trained in dietary assessment. The New Zealand food composition data NZ FOODfiles 2016 (Plant and Food Research Ltd., Palmerston North, New Zealand) will be used to estimate energy and nutrient intake and the data will also be adjusted for day-to-day variations using established procedures.

### 3.4. Physical Activity and Sleep Assessment

Participants were provided with an accelerometer, ActiGraph wGT3X-BT, to be worn on their non-dominant wrist for seven consecutive days, one week prior to baseline (day-7) and one week prior to week 12, to assess habitual physical activity and sleep patterns. The accelerometers were initialized to record data at a sampling frequency of 30 Hz in three axes: vertical, mediolateral, and anteroposterior [71], using ActiLife software (V6.13.3 Lite Edition, ActiGraph, FL, USA). Participants were asked to keep the accelerometer on at all times except during water-based activities such as showering, bathing, and swimming. They were also given a physical activity and sleep diary for recording instances when they removed the accelerometer for more than 5 min, activities that may not be tracked accurately by the accelerometer (e.g., swimming, biking, weight training), along with their respective intensities as well as sleep-related information such as the time they tried to go to sleep, the time they woke up, and the time they got out of bed. Days that consist of a minimum of 10 h wear time during waking hours or a minimum of 20 h wear time with imputed physical activity (physical activity recorded in the diary) and at least 2 h of sleep will be considered valid and data sets containing 3 or more valid days will be used for analysis [72].

### 3.5. Clinical Measurements

#### 3.5.1. Anthropometry

At the screening visit, the participants’ height and weight were measured to calculate body mass index (BMI) as weight in kilograms per height in meters squared. At the baseline visit (week 0) and final visit (week 12), the participant’s height, weight and waist circumference were measured to monitor any changes in weight and central adiposity with the addition of 500 mL milk each day.

All measurements were taken according to guidelines/protocols set out by the New Zealand Ministry of Health [73]. Briefly, height was measured to the nearest millimeter using calibrated stadiometers, weight was measured to the nearest 0.01 kg using calibrated scales and waist circumference was measured at the narrowest point between the lower costal border and the top of the iliac crest using anthropometric tape measures. All measurements were taken with shoes removed and light clothing. All anthropometric measurements were taken in duplicate, with a third measurement taken should the first two differ by 10% and the average value taken.

#### 3.5.2. Body Composition

A whole-body DEXA scan was performed at baseline and week 12 to assess body composition (fat mass and lean body mass) to support anthropometric measurements in the event of body weight changes. Dunedin participants were scanned on the Lunar iDXA scanner (GE HealthCare, Madison, WI, USA) while Christchurch participants were scanned on the Lunar Prodigy scanner (GE HealthCare, Madison, WI, USA), In preparation for the scan, participants were asked to remove any metal artifacts (e.g., jewelry, clothing with metal fasteners) from their person and keep on light clothing or be given a hospital gown to wear. They were positioned on the scanning table in a supine position with their arms by their side, parallel to but not touching the body, palms facing their legs and legs fully extended [74]. Offset scanning was performed for participants who did not fit in the scanning field of view, as per the Official Position of the ISCD [74].

#### 3.5.3. Blood Pressure

Blood pressure was taken at baseline visit (week 0) and final visit (week 12) according to the guidelines set by the Australian Expert Consensus [75] using an automated blood pressure measurement system with an appropriately sized upper arm cuff. Participants were seated with their back and arm supported in a relaxed positioned, legs uncrossed, feet flat on the floor, and the cuff at heart level. Their blood pressure was measured 3 times, and the average value taken.

#### 3.5.4. Functional Tests

A timed 5 m timed walk test was performed at baseline visit (week 0) and final visit (week 12) to assess functional mobility. Briefly, a pathway measuring 7 m was measured and marked at 0 m, 5 m, and 7 m with bright tape. Participants were instructed to stand behind the 0 m mark with the tip of their shoes just touching the mark and given a cue (“1, 2, 3, go”) to start walking from the 0 m mark to the 7 m mark without assistance or with assistive devices if necessary (which was consistent and documented from test to test) [76]. A researcher started the timer when the toes of the leading foot crossed the 0 m mark and stopped the timer when the toes of the leading foot crossed the 5 m mark. The test was repeated 3 times, and the average value was taken.

A handgrip strength test was performed at baseline and week 12 using a Jamar Plus+ Digital Hand Dynamometer (Performance Health, Warrenville, IL, USA). Participants were seated with their elbow by their side and flexed to 90 degrees and asked to hold on to the dynamometer set at position 2 from the inside with their dominant hand [77]. They were asked to squeeze the dynamometer as hard as possible and three successive measurements recorded to the nearest 1 kg were taken with 10–20 s rest between subsequent measurements, and the average value taken [77].

### 3.6. Biological Measurements

#### 3.6.1. Blood Samples

During the baseline and final visits, fasting blood samples were collected from participants by trained phlebotomists into two BD Vacutainer^®^ serum tubes for trace element determination with K_2_EDTA, two BD Vacutainer^®^ lithium-heparin tubes, and one BD Vacutainer^®^ K_2_EDTA tube (Becton, Dickinson and Company, Franklin Lakes, NJ, USA). Postprandial blood samples were collected from participants who took part in the sub-study into one BD Vacutainer^®^ lithium-heparin tube, one BD Vacutainer^®^ K_2_EDTA tube and one BD Vacutainer^®^ P800 tube (Becton, Dickinson and Company, Franklin Lakes, NJ, USA). The collected samples were kept on ice and processed within an hour. After aliquots of whole blood were taken from the lithium heparin tubes used in the main study, all vacutainers were centrifuged at 1500× *g* for 15 min at 4 °C to separate plasma from cells. Aliquots of both whole blood and plasma were then frozen at −80 °C until analysis.

Amino acids

Amino acid concentration in plasma will be analyzed as previously described [78] to examine the impact of sustained milk consumption on circulating amino acids by reverse-phase chromatography and fluorescence detection [79]. Briefly, the frozen plasma samples will be thawed, mixed with a 0.02% sodium azide solution, and centrifuged in 3000 MWCO Vivaspin 500 centrifugal concentrators (Sartorius, Goettingen, Germany) at 13,675× *g* for 40 min at 10 °C. The resulting samples will be mixed with norvaline as an internal standard solution and analyzed using an Agilent 1290 Infinity II HPLC system (Agilent Technologies Inc., Santa Clara, CA, USA) with automated online precolumn o-phthalaldehyde derivatization. Precolumn-derivatized free amino acids in solution will be separated on a C18 reversed-phase column and detected on a fluorescence detector set at 230 nm excitation wavelength with a 450 nm emission cutoff filter.

2.Blood lipids and Hemoglobin A1c (HbA1c)

Blood lipid (total cholesterol, high-density lipoprotein, triglycerides) and HbA1c concentrations in plasma will be analyzed to monitor potential cardiometabolic effects from increased milk consumption using a Cobas c311 Analyzer (Roche, Basel, Switzerland) according to manufacturer’s instructions. Briefly, the frozen plasma samples will be thawed, homogenized and centrifuged at 2000× *g* for 10 min before the samples are loaded into the analyzer with the appropriate reagent kits. Quality control will be performed using PreciControl ClinChem Multi 1 and 2 (Roche, Basel, Switzerland), and calibration will be carried out using Calibrator F.A.S. Proteins (Roche, Basel, Switzerland) and pooled serum. Low-density lipoprotein concentration will be calculated using the Friedwald equation [80].

3.Trace elements

Trace elements (zinc, selenium, and magnesium) concentration in serum will be analyzed using inductively coupled plasma mass spectrometry (ICP-MS) [81] to give objective markers of dietary intake and to assess changes in nutrient intake with the addition of milk to the diet. Briefly, 100 μL of thawed serum will be mixed with 200 μL of 65% nitric acid and 100 μL of hydrogen peroxide and vortexed briefly. The mixture will be incubated at 60 °C for 90 min before being cooled down with the addition of 2100 μL ultrapure water. The samples will be then vortexed and centrifuged for 3 min at 595× *g* before being loaded into an Agilent 7500ce ICP-MS (Agilent Technologies Inc., Santa Clara, CA, USA) for analysis as previously described [82].

4.Inflammatory markers

Inflammatory markers (high sensitivity C-reactive protein and α-acid-1-glycoprotein) in plasma will be analyzed to adjust for nutrient biomarkers that are altered in the presence of inflammation (e.g., serum zinc and selenium) [82] and as potential confounders of skeletal muscle strength and muscle mass [83]. Analyses will be undertaken using a Cobas c311 Analyzer (Roche, Basel, Switzerland) according to the manufacturer’s instructions. Briefly, the frozen plasma samples will be thawed, homogenized and centrifuged at 2000× *g* for 10 min before the samples are loaded into the analyzer with the appropriate reagent kits. Quality control will be performed using PreciControl ClinChem Multi 1 and 2 (Roche, Basel, Switzerland), and calibration will be carried out using Calibrator F.A.S. Proteins (Roche, Basel, Switzerland) and pooled serum.

5.Parathyroid hormone and vitamin D

Parathyroid hormone and vitamin D in plasma will be analyzed as they are potential confounders of skeletal muscle function [84] using Cobas e411 Analyzer (Roche, Basel, Switzerland) according to the manufacturer’s instructions. Briefly, the frozen plasma samples will be thawed, homogenized and centrifuged at 2000× *g* for 10 min before the sample is loaded into the analyzer with the appropriate reagent kits (Roche, Basel, Switzerland). Quality control for parathyroid hormone will be performed using PreciControl Varia (Roche, Basel, Switzerland), and calibration carried out using CalSet PTH (Roche, Basel, Switzerland) and pooled serum. Quality control for vitamin D will be performed using Vitamin D total II CalSet (Roche, Basel, Switzerland), and calibration carried out using PreciControl Vitamin D total II (Roche, Basel, Switzerland) and pooled serum.

6.Non-polar metabolites

The relative intensity of non-polar metabolites in plasma will be analyzed to identify lipid metabolites that are produced in response to the different study milks. Non-polar metabolites will be extracted from plasma as previously described [85]. Briefly, 10 μL of plasma will be mixed with 100 μL of butanol/methanol (1:1) with 10 mM ammonium formate containing a mixture of internal standards. The samples will be vortex mixed and incubated in a sonicator bath maintained at room temperature for 60 min before being centrifuged at 14,000× *g* for 10 min at 20 °C. The supernatant will then be transferred into sample vials with glass inserts for analysis. For analysis, the samples will be injected into a C18 reverse-phase column in a Shimadzu Nexera-x2 Ultra Performance Liquid Chromatography^®^ system equipped with a Shimadzu LCMS-9030 mass spectrometer (Shimadzu Corporation, Kyoto, Japan) [86]. The mass spectrometer will be operated in positive ionization mode to measure full MS1 spectra from 250 to 1250 *m*/*z* across the entire chromatogram and collect data independent acquisition (DIA) data in 20 *m*/*z* windows from 300 to 1100 *m*/*z*.

#### 3.6.2. Stool Samples

One day prior to their baseline and final study visits, participants collected stool samples using the OMNIgene GUT collection systems OMR205 and ME200. They kept the tubes at room temperature until delivery to the clinics the following day. The stool samples were vortexed, aliquoted into cryovials and stored at −80 °C until analysis.

DNA extraction from stool samples collected in OMR-205 was performed using the ZymoBIOMICS DNA/RNA Miniprep Kit (Zymo Research, Irvine, CA, USA) according to the manufacturer’s instructions. The quality and concentration of the extracted DNA were assessed on an Implen N60 NanoPhotometer (Implen GmbH, Munich, Germany) and visualized on a 1% agarose gel. Libraries for metagenomic sequencing will be prepared, and DNA will be sequenced by Annoroad Gene Technology Limited (Beijing, China).

The raw sequencing data will undergo quality check using FastQC (V0.11.9) [87] followed by preprocessing using Trimmomatic (V0.36) to remove adapter sequences, low-quality (Phred scores < 30), and short (<36 bp) sequencing reads [88]. Host contamination reads will be removed using “bbduk” from BBMap (V39.01) [89]. Read pairs will be joined using PEAR (V0.9.6) at the default setting [90] and any read pairs not joined by PEAR will be concatenated with a string of 50 N’s using the “fuse” function of the BBMAP package (V38.22-0) [89]. Joined and fused reads from different lanes of the same sample will be compiled into a final “clean” read sample file. HUMAnN (V3) will be used to identify and profile the functional potential of microbial communities in the samples [91]. Briefly, known microbial species will first be identified with MetaPhlAn2 [92]. A database of the samples will then be constructed by merging preconstructed, functionally annotated pangenomes through nucleotide mapping [93]. Finally, a translated search for reads not aligned to pangenomes will be performed using the UniRef90 database [94].

Metabolites from stool samples collected in ME-200 will be processed and analyzed as previously described [95].

### 3.7. Acute Sub-Study

During the baseline and final study visits, participants who consented to the sub-study consumed a bolus dose (500 mL) of their allocated milk. Sequential blood samples were drawn at 0, 60, 120, 180, 240, and 300 min to measure circulating levels of amino acids, insulin, and triglycerides.

Amino acid concentrations in plasma will be analyzed as described in Section 3.6.1. to assess the impacts of sustained milk consumption on circulating amino acid appearance. Circulating insulin and triglycerides will be assessed to determine the GlucoTRIG index of the study milks, a novel index for classifying the overall healthiness of meals based on the glycemic and lipemic responses they induce [56]. Insulin and triglycerides in plasma will be analyzed using Cobas e411 Analyzer (Roche, Basel, Switzerland) and Cobas c311 Analyzer (Roche, Basel, Switzerland), respectively, according to the manufacturer’s instructions. Briefly, the frozen plasma samples will be thawed, homogenized and centrifuged at 2000× *g* for 10 min before the sample is loaded into the analyzers with the appropriate reagent kits. Quality control for insulin will be performed using PreciControl Multimarker (Roche, Basel, Switzerland) and calibration carried out using Insulin CalSet (Roche, Basel, Switzerland) and pooled serum. Quality control for triglycerides will be performed using PreciControl ClinChem Multi 1 and 2 (Roche, Basel, Switzerland) and calibration carried out using Calibrator f.a.s. (Roche, Basel, Switzerland) and pooled serum.

During these timepoints, participants also completed a questionnaire to self-report appetite and satiety. This questionnaire, comprising eight 100 mm Visual Analog Scales, consists of questions relating to feelings of perceived hunger, satisfaction, fullness, and desire for specific food types [96]. This allows the study of potential adaptations to daily milk consumption.

## 4. Data Analysis

### 4.1. Sample Size Calculation

To date, no study has compared the impacts of sustained bovine, caprine and ovine milk consumption on digestive comfort in older adults. Thus, sample size calculation was based on the pooled mean (SD) of constipation pain domain scores in the GSRS in healthy people [97] and those with gastroesophageal reflux disease [98]. A sample size of 34 women per group will detect a minimal important difference (MID) of 0.6 [64] within the abdominal pain domain of the GSRS from baseline to week 12 between each intervention group and the control group, with 5% significance (alpha) and 80% power (beta = 0.20). This sample size will also detect a difference in mean nutrient intake (protein, fat, calcium, magnesium) between each milk group compared with the control group assuming effect sizes of 0.7. The sample was increased to 40 participants per group to allow for uncertainty of the estimates and a predicted attrition rate of 15%.

For the acute sub-study, a sample size of 25 women per group will detect a 20% difference in plasma leucine area under the curve appearance at 60 min post-ingestion between bovine milk and ovine milk [52] (no data for caprine-bovine comparison), with 5% significance (alpha) and 80% power (beta = 0.20).

### 4.2. Statistical Analysis

All data collected will be analyzed for both intention-to-treat (ITT) and per-protocol (PP) populations. The ITT population will include all randomized participants who have taken at least one serving of their assigned study milk whereas the PP population will include all randomized participants who have completed the study and have complied with at least 80% of the intervention without major protocol violations. All analyses will be undertaken using SPSS v28 (IBM, New York, NY, USA) or R version 4.3.3 (The R Foundation, Vienna, Austria) by blinded researchers with guidance from an independent biostatistician and statistical significance will be set at a two-tailed *p* value of less than 0.05.

All study outcomes will be summarized using descriptive statistics such as mean, standard deviation, median, range, frequency and percentage according to data type. Univariate and covariate analyses will be performed to evaluate and compare the changes for patient reported outcomes, blood biomarkers, blood pressure, anthropometric measurements, body composition, and functional tests. Chi square tests and logistic regression will be used to analyze categorical data. One-way ANOVA and general linear mixed models with ‘participant’ as a random factor, ‘study timepoint’ and ‘study allocation’ as fixed factors will be used to analyze continuous data. Further models incorporating baseline levels and other potential confounding or explanatory variables may also be used to explore the outcomes and the relationships between specific outcomes.

For stool microbiome analysis, the relative abundance of bacterial taxa, gene abundance, and alpha and beta-diversities will be compared. If these data do not meet the assumptions for parametric testing, the non-parametric permutation ANOVA known as ANCOMBC [99] will be used to analyze microbiome data to account for the high dimensionality, non-normality, and phylogenetic structure of the data.

A priori planned comparisons of within-group and between-group changes will focus on the changes from baseline to the week 12 endpoint.

## 5. Data Management

### 5.1. Privacy and Confidentiality

Researchers responsible for data collection underwent training in Good Clinical Practice (GCP) to ensure the privacy and confidentiality of all study participants and data collected. No identifying or identifiable information about participants, including names, dates of birth, images, or any identifying details, will be disclosed in any reports or publications resulting from this study. Identifying information of participants was collected at enrolment, entered, and can only be accessed via REDCap (Nashville, TN, USA) by these researchers. Participants were allocated unique identifiers at enrolment which were used on all study materials in lieu of any identifying information.

### 5.2. Data Quality and Assurance

Participants were encouraged to complete most study questionnaires electronically, either via email links or the study app to minimize the risk of errors during data entry. Those who preferred paper questionnaires were accommodated and any paper-based data were entered into the REDcap data collection system by the research team.

For the entry of raw diet diary data into FoodWorks11 (Xyris, Brisbane, Australia), a data entry assumptions spreadsheet will be set up to ensure consistency in data entry and the research team will independently audit 20% of the diet diaries entered into FoodWorks. Extreme values for specific nutrients will be identified and clarified by reviewing the spreadsheet and diet diaries.

No formal data monitoring committee was set up as the interventions in the study are considered low risk, and risks beyond those of standard care and everyday life were anticipated. Data monitoring took place during research team meetings.

### 5.3. Data Storage

Data files for biological data will be stored in Excel for raw data, and as data analysis files in SPSS or R. These files will only be identifiable via the participants’ unique identifiers. Biological samples will be stored in secure facilities with limited access until the publication of study results but not longer than 10 years. The samples will be destroyed hygienically thereafter following NZS 4304:2002 [100] Management of Healthcare Waste or with the appropriate karakia for Māori participants.

Paper-based data files will be stored in locked filing cabinets in locked offices. Electronic data files will be stored on a password-secured University of Otago server or Otago OneDrive cloud storage and will be accessed and downloaded, as the need arises, to the password-protected computers of the research team stored on locked premises.

The principal investigator will retain all data concerning the study for at least 15 years, and participants can exercise access rights at any time.

All data will undergo an automatic anonymization process before analysis.

## 6. Expected Results and Contributions

This study is the first to investigate and compare the health impacts of consuming different ruminant milks in older women over a sustained period of time. As digestive discomfort from bovine milk consumption is a significant barrier to milk consumption [101,102], the study will shed light on how compositional and functional differences between bovine, caprine, and ovine milks [34,39,40,41] impact self-reported digestive comfort and bowel habits, and the gut microbiota. It will also highlight how the addition of two cups of ruminant milk each day influences the nutritional status of community-dwelling older women, a group often at higher risk of nutritional insufficiency [32,103]. Another significant barrier to milk consumption is concerns over cardiometabolic health [104,105]. Findings from observational and randomized controlled trials largely suggest a neutral or inverse association between bovine milk and dairy product intake and cardiometabolic outcomes or biomarkers [38,106,107,108]. While observational studies generally include a broader age range, most randomized controlled trials focus on healthy individuals or those with metabolic abnormalities from the general adult population (age 18–65). Hence, our study will add to existing research by exploring how different milks impact cardiometabolic biomarkers in healthy older women.

Furthermore, digestive comfort and the gut microbiota have been linked to psychological wellbeing and sleep [4,109,110,111,112,113,114]. While milk’s potential impacts on sleep and mood have been explored, the type of milk studied is often not specified and human studies are limited and inconclusive, especially in older adults [115,116]. Our study expands on this by investigating how different milks affect sleep quality and psychological distress in older women over time and whether these factors have any bidirectional relationships with changes in digestive comfort and the gut microbiota.

In addition, bovine milk has been studied extensively for its ability to stimulate muscle protein synthesis [117,118], a process critical for preserving muscle mass in older adults who often experience sarcopenia, the progressive loss of muscle mass and function with age [119]. While findings from previous studies are largely promising, they tend to focus on isolated milk proteins rather than whole dairy products [118,120]. In our study, we will not directly assess the effects of milk consumption on muscle protein synthesis but will explore the impacts of incorporating whole milks in the daily diets of older women on their body composition, including lean muscle mass, physical function, and physical activity levels.

Finally, this study will explore the sensory attributes of and emotional responses to different ruminant milks. Various factors influence an individual’s motivation to eat, including the sensory properties of the food as well as the emotions evoked by the food [121]. Consumer research has shown that the distinct flavor profiles of caprine and ovine milk can evoke strong emotions which serve as either motivators or deterrents for consumption [122,123]. While previous studies have mostly concentrated on the general adult population, our study will provide important insights into the experiences of healthy older adults, for whom pleasure is a key motivator for eating well [124].

The acute sub-study would have allowed for the assessment of acute postprandial responses to consuming the study milks and how these responses might change over 12 weeks of sustained milk consumption. However, logistical constraints limited recruitment into the sub-study, and the required participant numbers for each milk group, as per the power calculations, could not be met. Nevertheless, the findings of the main study will help to inform nutrition recommendations and interventions to meet the dietary needs of this expanding population group and facilitate the development of targeted functional food products.

## Figures and Tables

**Figure 1 nutrients-16-04215-f001:**
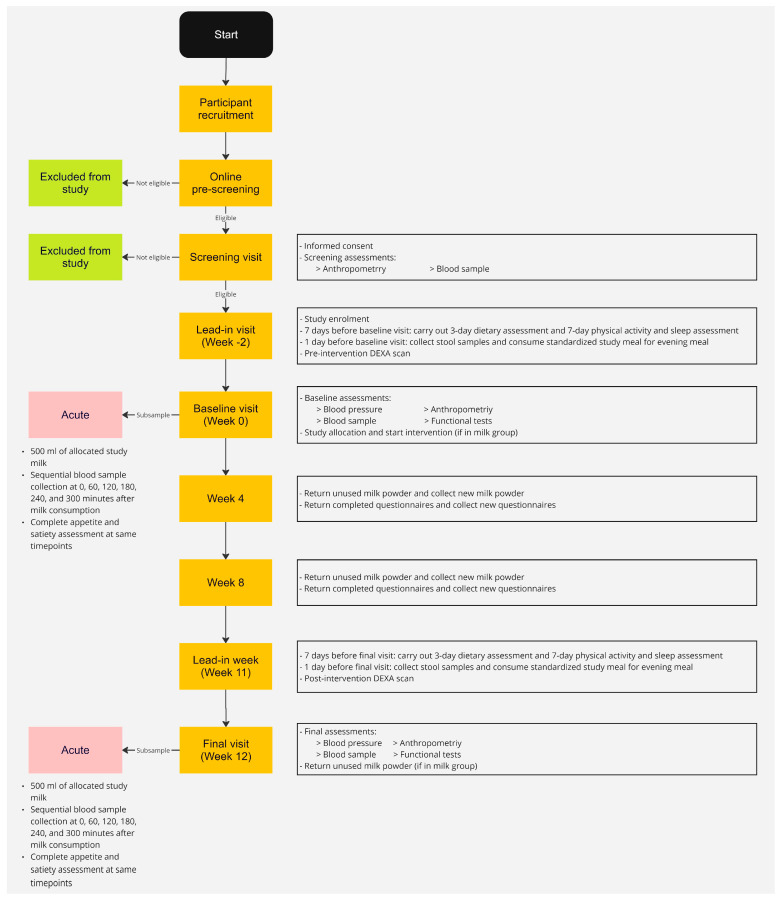
Flow diagram of participant journey.

**Table 1 nutrients-16-04215-t001:** Nutrient Composition of Study Milk Powder *.

	Milk Powder Dose	Protein Content		Fat Content	
	g/day	g/100 g	g/day	g/100 g	g/day
Bovine	70	26.7	18.69	23.6	16.52
Caprine	70	31.8	22.26	26.2	18.34
Ovine	88	33.3	29.304	29.9	26.312

* Reference [62].

## Data Availability

The original contributions presented in the study are included in the article, further inquiries can be directed to the corresponding authors.

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
