# Peer review of "Study Protocol for a Randomized Controlled Trial Investigating the Effects of the Daily Consumption of Ruminant Milk on Digestive Comfort and Nutrition in Older Women: The YUMMI Study"

_nutrients, 2024, doi:10.3390/nu16234215_

Round 1
Reviewer 1 Report
Comments and Suggestions for Authors
Overall, I really liked the protocol as appropriate detail and syntax was provided. I do have a few specific comments.
line 62, 67, and others: I suggest that authors use 'because of' rather than 'due to' within the text throughout the manuscript.
line 185-187: Authors need to describe criteria for allocating participants to the acute sub-study.
line 230: I question the wisdom of leaving the stool samples set at room temperature for variable times post collection. End products and microbial population will likely change during the fermentation post collection.
Table 1: Please explain why you offer more ovine milk powder to participants than the bovine and c aprine milk powders.
line 329: You state that the weekly questionnaires follow. I do not see the questionnaire in the following text. Please modify appropriately.
lines 371,416, 419, and 449: Please remove the vague and all-inclusive 'etc' here and throughout the manuscript.
lines 324-328: At first, I thought that this detail is not needed when the procedure is references. This statement applies for several of the following paragraphs as well. Please reconsider the necessity of the procedural details that follows you Briefly sentences. Be sure to convert rpm to x g here and elsewhere.
Comment: This extensive experiment will provide a great amount of data regarding the testing of three different milks on numerous variables. I hope you are providing enough milk to the people to have an impact during the 12-week experiment. Also, I presume that you have sufficient funds to complete all the assays that you plan. Do you really believe that the highly digestible milks will have an impact on the microbial population of feces? Congratulations on a well-written manuscript.
Reviewer 2 Report
Comments and Suggestions for Authors
The manuscript is well-written and understandable.
The research topic and the specific study ideas are relevant for conceptual research and also for nutritional practice.
The authors may want to clarify why only older women were enrolled?
Do the authors not expect relevance for men and/or younger age groups too?
The exact randomization procedure could be explained.
How did the authors assure compliance to the intervention in detail?
The study time frame is 12 weeks only. How conclusions can be made about long-term intake? Via an extrapolation?
How can the findings be implemented in daily-life consumption? People often follow their beliefs and convictions, it is not easy to introduce changes in nutritional behaviors.
Will costs for those milk alternatives be taken into account? They may be relevant for decision making.
Will the background, i.e. past milk consumption behavior be taken into account. It can lead to difficulties for those that need to change their intake due to the intervention, compared to those who already have a preference for milk alternatives.
Round 2
Reviewer 1 Report
Comments and Suggestions for Authors
I find your responses to suggestions to be acceptable.